# Interaction between Metabolic Genetic Risk Score and Dietary Fatty Acid Intake on Central Obesity in a Ghanaian Population

**DOI:** 10.3390/nu12071906

**Published:** 2020-06-27

**Authors:** Sooad Alsulami, David A. Nyakotey, Kamila Dudek, Abdul-Malik Bawah, Julie A. Lovegrove, Reginald A. Annan, Basma Ellahi, Karani Santhanakrishnan Vimaleswaran

**Affiliations:** 1Hugh Sinclair Unit of Human Nutrition, Department of Food and Nutritional Sciences, University of Reading, Reading RG6 6AP, UK; s.alsulami@pgr.reading.ac.uk (S.A.); k.j.dudek@student.reading.ac.uk (K.D.); j.a.lovegrove@reading.ac.uk (J.A.L.); 2Department of Clinical Nutrition, Faculty of Applied Medical Sciences, King Abdulaziz University, Jeddah 21589, Saudi Arabia; 3Department of Biochemistry and Biotechnology, College of Science, Kwame Nkrumah University of Science and Technology, Accra GH233, Ghana; dnyakotey@gmail.com (D.A.N.); malikbawa2008@yahoo.com (A.-M.B.); reggie@imtf.org (R.A.A.); 4Hugh Sinclair Unit of Human Nutrition and Institute for Cardiovascular and Metabolic Research, University of Reading, Reading RG6 6AP, UK; 5Faculty of Health and Social Care, University of Chester, Riverside Campus, Chester CH1 4BJ, UK

**Keywords:** genetic risk score, obesity, Ghana, GONG, fat intake, gene–diet interaction

## Abstract

Obesity is a multifactorial condition arising from the interaction between genetic and lifestyle factors. We aimed to assess the impact of lifestyle and genetic factors on obesity-related traits in 302 healthy Ghanaian adults. Dietary intake and physical activity were assessed using a 3 day repeated 24 h dietary recall and global physical activity questionnaire, respectively. Twelve single nucleotide polymorphisms (SNPs) were used to construct 4-SNP, 8-SNP and 12-SNP genetic risk scores (GRSs). The 4-SNP GRS showed significant interactions with dietary fat intakes on waist circumference (WC) (Total fat, P_interaction_ = 0.01; saturated fatty acids (SFA), P_interaction_ = 0.02; polyunsaturated fatty acids (PUFA), P_interaction_ = 0.01 and monounsaturated fatty acids (MUFA), P_interaction_ = 0.01). Among individuals with higher intakes of total fat (>47 g/d), SFA (>14 g/d), PUFA (>16 g/d) and MUFA (>16 g/d), individuals with ≥3 risk alleles had a significantly higher WC compared to those with <3 risk alleles. This is the first study of its kind in this population, suggesting that a higher consumption of dietary fatty acid may have the potential to increase the genetic susceptibility of becoming centrally obese. These results support the general dietary recommendations to decrease the intakes of total fat and SFA, to reduce the risk of obesity, particularly in individuals with a higher genetic predisposition to central obesity.

## 1. Introduction

Obesity is a known risk factor for several health conditions, including type 2 diabetes, cardiovascular diseases, hypertension and cancer, and hence it is considered as an increasing public health problem worldwide, including in Africa [1,2]. Obesity prevalence varies widely between African countries with a range of 5.3% in Uganda to 30% in Nigeria and 45.7% in South Africa [2]. A recent systematic review has reported that nearly 43% of Ghanaian adults are either overweight or obese and that the prevalence of overweight and obesity was higher in women and urban dwellers [3]. While obesity is strongly affected by changes in environmental factors (such as dietary intake, sedentary lifestyle, and urbanization), the composition of the gut microbiome, the disruption of circadian rhythms, exposure to endocrine-disrupting chemicals and epigenetic modifications [4,5,6,7,8,9], it also has strong genetic determinants with a heritability rate from 40 to 70% [10,11]. Genome-wide association studies (GWAS) in European populations have revealed more than 100 loci to be associated with the body mass index (BMI) [12,13,14,15,16,17,18]. However, these genetic associations have not been consistently replicated in African populations [19,20,21,22,23], which could be attributed to differences in lifestyle and genetic architecture [24].

Given that single nucleotide polymorphisms (SNPs) have relatively small effect sizes on obesity, several studies have aggregated information from multiple-risk variants into a polygenic genetic risk score (GRS) [13,15,25,26,27,28,29]. Employing a combined risk allele score is an efficient and effective approach in maximising statistical power, decreasing the drawback of multiple testing, and widening the generalisable nature of genetic associations [25,27]. A study among a rural population of Gambia demonstrated a positive association between a GRS of 28 SNPs and BMI and adult weight, whereas no association was found with the single SNP analysis [30,31]. Although genetic research in Africans is increasing in numbers [22], only a few studies have examined the association of GRS with obesity in Africa [30,32,33], which highlights the need for further research in African populations.

Current evidence has shown that heritability estimates for obesity-related traits can be modified by lifestyle factors such as diet and physical activity. Several studies have reported significant GRS–diet interactions on obesity-related traits. Studies in European populations have shown that the genetic association with BMI was stronger with higher intakes of sugar-sweetened beverages (SSBs) and fried foods than among those with lower intakes [34,35]. Studies have also shown that genetic associations with BMI in Europeans can be modified by the levels of physical activity, television watching, and changes in sleep pattern [36,37]. In addition, higher adherence to healthy eating patterns have shown to reduce BMI in Europeans despite having increased genetic susceptibility to obesity [38]. Gene–lifestyle interaction studies have largely been conducted in populations of European ancestry, and the replication of these studies in African populations remains unknown [36,39]. Therefore, our study aimed to investigate the association of GRS with obesity-related traits and to examine whether lifestyle factors such as dietary intake and physical activity modified these associations in the Ghanaian population.

## 2. Methods

### 2.1. Study Population

The Genetics of Obesity and Nutrition in Ghana (GONG) study is a cross-sectional study that was conducted in the Oforikrom Municipality in Kumasi, Ashanti region, Ghana. The GONG study was conducted as part of the ongoing GeNuIne (Gene–Nutrient Interactions) Collaboration, the main objective of which is to investigate the effect of gene–nutrient interactions (nutrigenetics) on metabolic disease outcomes using population-based studies from various ethnic groups [40,41]. The Oforikrom Municipal Assembly is one of the five Municipal Assemblies carved out of the Kumasi Metropolitan Assembly. There are seventeen recognized communities in this Municipal Assembly, with an estimated total population of 360,254. Five communities (Ayeduase, Bomso, Ayigya, Oforikrom and Kotei) were randomly selected from the list of communities in the Oforikrom Municipal Assembly. In each community, a central point was located (a vehicle station, marketplace or other landmarks). A fieldworker entered the first house that faced either North, South, East or West of that central point, and randomly recruited one respondent from each household. Upon exiting a house, the fieldworker entered the next house, and the house-level selection process was repeated.

Three hundred and two free-living and healthy (with no physical complaints or prior diagnosis of cardiometabolic disease) adult volunteers, both men and women, were screened and recruited for the study by trained researchers. The inclusion criteria included the following: healthy individuals aged 25 to 60 years old and being Asante (both parents must be Asante). The exclusion criteria included the following: participants less than 25 years old or older than 60 years, those with existing cardiovascular complications or disease, those with a previous history of hypertension, type 2 diabetes or cardiovascular diseases, participants with any communicable or non-communicable chronic diseases, pregnant women and participants on lipid-lowering drugs, anti-diabetic drugs or anti-hypertensive drugs. A medical screening questionnaire was developed to screen participants for inclusion or exclusion from the study.

This study was approved by the Council for Scientific and Industrial Research (CSIR) Institutional Review Board (IRB) (Ref: RPN 003/CSIR-IRB/2018). In addition, this study was approved by the Metro Director of Health Services, Kumasi (KMHD/MPHs/13). All participants signed informed consent prior to their participation.

### 2.2. Data Collection

Structured questionnaires were used to elicit information about the participants’ demographic characteristics, dietary intakes, physical activity levels, sleep and sunshine exposure patterns and medical history. Fieldworkers were trained before the start of data collection. Survey instruments were also pre-tested on the 10 July 2018 to enhance the field workers’ understanding of questionnaires, ensure clearness and avoid ambiguity. Data collection took place from July to September 2018.

### 2.3. Anthropometric Measurements

Height, weight, percentage of body fat and visceral fat, waist circumference (WC) and hip circumference (HC) were measured. The measurements were taken with respondents wearing light clothing. Height was measured with a stadiometer (Seca 213 mobile stadiometer, Hamburg, Germany) to the nearest 0.1 cm with participants standing upright without shoes. Weight was measured using an OMRON Body Composition Analyzer to the nearest 0.1 kg. The same equipment provided values for BMI, percentage of body fat and visceral fat. WC and HC measurements were taken using a non-extensible measuring tape with participants in light clothing. The WC was measured just above the naval to the nearest 0.1 cm whereas the HC was measured at the level of the greater trochanter to the nearest 0.1 cm. The waist-to-hip ratio (WHR) was calculated by dividing WC by HC.

### 2.4. Physical Activity and Dietary Assessments

The health-related physical activity level of participants was measured using the interviewer-administered Global Physical Activity Questionnaire (GPAQ) version 2 developed by the World Health Organization (WHO) for physical activity surveillance [42]. This questionnaire contains 16 questions (P1–P16) which gather information on the respondent’s engagement in physical activities under three domains or settings (work-related activity, transportation and recreational activities) as well as sedentary behaviours. The total physical activity per week was calculated in Metabolic Equivalents (MET- minutes) and the respondents who had total physical activity ≥ 600 MET- minutes/week were classified as active while those who had < 600 MET- minutes/week were classified as inactive [42].

A three-day repeated (two weekdays and one weekend) 24 h dietary recall method was used to elicit the information concerning the participants’ dietary intake. Participants were requested to recollect all the meals taken as well as the times of the meal consumption in the previous day. Common household measures were used to estimate the actual quantities of foods and drinks consumed by the participants. The nutritional composition of the foods eaten was then analysed using the Nutrient Analysis Template (Food Science and Nutrition Department, University of Ghana, Accra, Ghana, 2010).

### 2.5. SNP Selection

Fifteen SNPs near or in 8 obesity-susceptibility loci were chosen for the study based on the previous GWAS for metabolic traits [12,13,14,15,16,17,18]. These include Transcription factor 7-like 2 (*TCF7L2)* (rs12255372, rs7903146), melanocortin 4 Receptor (*MC4R*) (rs17782313, rs2229616)*,* fat mass and obesity-associated (*FTO*) (rs9939609, rs10163409)*,* adiponectin (*ADIPOQ*) (rs266729, rs17846866)*,* Potassium voltage-gated channel subfamily Q member *1(KCNQ1)* (rs2237892, rs2237895)*,* Cyclin dependent kinase inhibitor 2A/2B (*CDKN2A/2B*) (rs10811661), Calpain 10 (*CAPN10*) (rs3792267, rs5030952, rs2975760) and Peroxisome proliferator-activated receptor gamma (*PPARG*) (rs1801282). Three of these SNPs, *KCNQ1* (rs2237895), *ADIPOQ* (rs17846866) and *CAPN10* (rs2975760), reported significant deviations from Hardy–Weinberg Equilibrium (HWE) (*p* < 0.05) and were excluded from the current analysis. The detailed information of the 15 SNPs is shown in Appendix A.

### 2.6. Genotyping

Blood samples for the measurement of DNA were transported in dry ice to the United Kingdom (UK). Genomic DNA was extracted from a 5 mL whole blood sample from each participant and genotyping was performed at the LGC Genomics (http://www.lgcgroup.com/services/genotyping), which employs the competitive allele-specific PCR-KASP^®^ assay.

### 2.7. Construction of the Metabolic GRSs

To evaluate the combined effects of the 12 SNPs on obesity-related traits, an additive model was used to construct the unweighted metabolic GRSs (Figure 1). We did not weigh the risk alleles based on their individual effect sizes, because no previously reported effect sizes were available for these SNPs for the Ghanaian population, and it has been shown that the weighting of risk alleles may only have limited effects [43]. The unweighted metabolic GRSs were calculated by the summation of the number of risk alleles across the 12 variants. The risk alleles were defined as alleles previously associated with an increased risk of obesity in the literature. To reduce the bias caused by the missing data, only those participants without any missing data were included in our metabolic GRS analysis. Different metabolic GRSs were constructed including the 12-, 8- and the 4-SNP GRSs. The 12-SNP GRS included the following SNPs: *TCF7L2* (rs12255372, rs7903146), *MC4R* (rs17782313, rs2229616), *FTO* (rs9939609, rs10163409), *ADIPOQ* (rs266729), *KCNQ1* (rs2237892), *CDKN2A/2B* (rs10811661), *CAPN10* (rs3792267, rs5030952) and *PPARG* (rs1801282), and the score ranged from 0 to 9 risk alleles. In the 12-SNP GRS analysis, no significant results were identified which might be because four of the SNPs had a minor allele frequency (MAF) of less than 5%. Therefore, we excluded the four SNPs: *MC4R* (rs2229616)*, FTO* (rs10163409), *CDKN2B (rs10811661)* and *PPAR* (rs1801282) and created an 8-SNP GRS. No significant findings were observed using the 8-SNP GRS; this might be because four of the eight SNPs (*ADIPOQ (rs266729)*, *KCNQ1 (rs2237892)* and *CAPN10 (rs3792267, rs5030952)*) have not shown consistent associations with obesity-related traits in other populations [44,45,46,47,48,49]. Hence, these four SNPs were removed and a 4-SNP GRS including the SNPs (*TCF7L2* (rs12255372, rs7903146), *MC4R* (rs17782313), *FTO* (rs9939609)) that have shown consistent associations with obesity across several populations was constructed. The 4-SNP GRS ranged from 0 to 6 risk alleles and significant results were observed. Based on the median number of each GRS, the individuals were separated into two groups.

Given that there were no previously reported effect sizes available for these SNPs for the Ghanaian population, we were unable to perform sample size calculation.

Data analyses were performed using Statistical Package for the Social Sciences (SPSS) software (version 24; SPSS Inc., Chicago, IL, USA). A natural log transformation was used for the non-normally distributed variables. Unadjusted differences of descriptive characteristics between the overweight/obese and non-obese participants were calculated using an independent samples *t*-test for continuous variables. General linear models were used to examine the association between the metabolic GRSs and obesity traits. GRS–lifestyle interactions were analysed by including the interaction terms in these models. Models were adjusted for covariates including sex, age and BMI (when BMI is not an outcome). Nutrient–GRS interaction analysis was additionally adjusted for total energy intake. All GRS–lifestyle interactions reaching a nominal level of significance (*p* < 0.05) were investigated further using binary analysis. Based on the median intake of total fat—saturated fatty acids (SFA), monounsaturated fatty acids (MUFA), and polyunsaturated fatty acids (PUFA)—the individuals were separated into two groups: ‘’below the median group’’ and ‘’above the median group”. Within each group, the association between the GRS and the outcome was examined. We also tested for GRS–sex interactions to test if sex influenced the genetic associations with obesity traits. The lifestyle factors investigated in our study included physical activity and the total dietary intake of fat, protein, carbohydrate and fibre. Significant interactions between the GRS and the total fat intake were further investigated to examine the influence of fat subtypes including saturated fatty acids (SFA), monounsaturated fatty acids (MUFA), and polyunsaturated fatty acids (PUFA). Two-tailed value of *p* < 0.05 was considered statistically significant.

## 3. Results

### 3.1. Characteristics of the Study Participants

The anthropometric and dietary characteristics of the study participants are presented in Table 1. The mean age and BMI of the total sample were 38.17 ± 9.64 years and 26.63 ± 4.99 kg/m^2^, respectively. Overweight/obese individuals were older than the non-obese (*p* < 0.05). Moreover, the dietary intakes were significantly different between the two groups. Overweight/obese individuals reported significantly lower intakes of total calories, protein, carbohydrate, total fat, fibre, SFA, MUFA and PUFA compared to the non-obese (*p* < 0.05). Women had significantly higher levels of BMI, body fat percentage and WHR compared to men, despite the men consuming significantly higher levels of carbohydrate, protein and fat (*p* < 0.05) (Appendix A).

### 3.2. Effect of Metabolic GRSs on Obesity-Related Traits

We first investigated the combined effect of 12 common SNPs on obesity-related traits and no significant associations were observed (Appendix A). Similar results were found using an 8-SNP GRS (Appendix A) and a 4-SNP GRS (Table 2).

### 3.3. GRS–Lifestyle Interactions on Obesity-Related Traits

There was a significant interaction of the 4-SNP GRS with dietary fat intake (g/day) on WC (Total fat, P_interaction_ = 0.01; SFA, P_interaction_ = 0.02; PUFA, P_interaction_ = 0.01 and MUFA, P_interaction_ = 0.01, Table 3). Individuals with ≥3 risk alleles had a significantly higher WC compared to those with <3 risk alleles, among individuals with higher intakes of total fat (>47 g/day), SFA (>14 g/day), PUFA (>16 g/day) and MUFA (>16 g/day), (Figure 2a–d). There was also a significant interaction between 4-SNP GRS and dietary fibre intake (g/day) on body fat percentage (P_interaction_ = 0.04). Individuals with <3 risk alleles had a significantly lower body fat percentage compared to those with ≥3 risk alleles (*p* = 0.02), among individuals with a higher intake of fibre (>19 g/day). In addition, there was a significant interaction between the 4-SNP GRS and physical activity on WHR (P_interaction_ = 0.002). However, the finding was not significant after stratifying them by physical activity levels. Some significant interactions were observed between the 12- and the 8-SNP GRSs and lifestyle factors on obesity-related traits (Appendix A), however, none of these interactions were significant after binary analysis. Given the significant differences in the dietary intakes and obesity-related outcomes between men and women, interactions between the 4-SNP GRS and sex were tested but no significant results were found (Appendix A).

## 4. Discussion

To our knowledge, this is the first nutrigenetic study investigating the interaction between metabolic GRSs and lifestyle factors on obesity-related traits in a Ghanaian population. Our study provides evidence for an interaction between the 4-SNP GRS and fat intake on WC, where individuals with ≥3 risk alleles had a significantly higher WC compared to those with <3 risk alleles among those who consumed a diet high in total fat, SFA, MUFA and PUFA. These results are in accordance with the general dietary recommendations, which suggest that the population decrease their intakes of total fat and SFA, to reduce the risk of obesity, and this will be more applicable in individuals with a higher genetic predisposition to obesity. Our findings are of importance to public health, considering the high consumption of foods that are rich in SFA and MUFA in the Ghanaian population [50].

Our study is the first study of its kind, investigating the effect of different metabolic GRSs (the 12-, 8- and the 4-SNP GRS) on obesity-related traits in a Ghanaian population. We found that none of the metabolic GRSs were significantly associated with obesity-related traits in the Ghanaian population, which contradicts the findings of the previous GRS-related studies in European and African populations [15,25,26,27,28,29,30,32,33]. Efforts to replicate previously reported genetic associations of individual SNPs with obesity measures in non-African populations have shown limited success among Africans [23,31,51,52], which is also in line with the findings from the present study. Several factors are likely to be involved in such discrepancies between our findings and genetic association studies in Europeans. First, the metabolic GRS in the present study was constructed based on variants strongly associated with BMI in European populations, which raises the question of the usefulness, applicability and accuracy of using this metabolic GRS in our African population. Analysing the genetic associations of such variants with obesity-related traits in African population may not be ideal because of differences in risk allele frequency and effect size across populations [53,54]. Second, the ‘lead’ SNPs identified in Europeans might tag smaller regions in Africans [19,20,55] and the ‘true’ causal polymorphisms might be at different loci [56]. A systematic review of genetic research in African samples has reported that more than 300 SNPs in 42 loci analysed in relation to obesity, but only a few positive associations were replicable in Africans [57]. Of the 36 variants previously established by GWAS in non-African populations, only the SNPs located at the *FTO* and *MC4R* loci were significantly associated with obesity in Nigerians, Ghanaians and black South Africans [58,59]. Furthermore, in a large-scale GWAS meta-analysis consisting of 71,412 individuals of African ancestry, of the 36 previously identified BMI-associated SNPs in Europeans, only five variants reached a genome-wide significant level in Africans [60]. Such inconsistencies in results are likely due, in part, to the variation in the genetic architecture between populations of different ancestry [61]. African populations are characterised by greater genetic variation, reduced patterns of linkage disequilibrium (LD) and more haplotype diversity in comparison with populations of another ancestry, which may cause difficulties in replicating previously reported genetic associations [61]. Hence, future studies with a larger sample size are needed to investigate the combined effect of a larger number of genetic variants on obesity-related traits in the Ghanaian population.

Our study has identified significant interactions between the 4-SNP GRS and intakes of total fat, SFA, PUFA and MUFA on WC, an indicator of central obesity that has been associated with the increased risk of morbidity and mortality [62,63]. Our findings suggest that dietary fatty acid consumption and composition may have the potential to influence the genetic susceptibility of becoming centrally obese. Evidence is limited concerning the GRS–diet interactions on obesity and its related traits, and most of the research has focused on the influence of a single locus [64,65,66], despite the genetic effects on obesity being polygenic [13]. Our results are consistent with previous findings generated from single-locus gene–diet interactions on obesity, in which fat intake is considered as an important lifestyle modulator of genetic associations with obesity-related traits. Two previous studies in 2163 participants from two independent United States (US) populations and in 28,449 individuals living in Malmö have shown significant interactions of the *FTO* SNP rs9939609 with total dietary fat on BMI [64,67], however, a large-scale meta-analysis of 177,330 individuals (154,439 Whites, 5776 African Americans and 17,115 Asians) failed to identify this interaction [68]. In addition, studies in 2163 participants from two US populations, 1754 French individuals and 354 Spanish children and adolescents have demonstrated a significant interaction of *FTO* SNP rs9939609 with SFAs [64,65,66] and MUFAs [64] on BMI. Furthermore, a study in 305 obese individuals in Finland reported that the high intake of MUFA was associated with weight loss among carriers of the risk allele (A) *FTO* rs9939609 [69]. Additionally, a study in 1680 South Asians has shown a significant interaction of the risk allele ‘T’ of the *TCF7L2* SNP rs12255372 with fat intake on high-density lipoprotein cholesterol (HDL-C) [70]. Studies on GRS–diet interactions on obesity traits have mainly focused on European populations [71,72,73]. In agreement with our study, data from UK Biobank [72] and two studies from the US [71] have reported significant interactions between the GRS and dietary intakes of total fat and SFA on WC; the GRS was associated with a higher WC among individuals with high intakes of total fat and SFA. However, the interactions on BMI were not identified in the present study, which contradicts the previously reported findings [71,72]. Hence, larger studies are required to replicate our GRS–fat intake interactions on WC in the Ghanaian population.

Several studies have investigated the impact of dietary fat on obesity measures; however, the findings have been inconsistent [74]. For instance, prospective studies have examined the relationship between the intake of long-chain omega-3 (LC n−3)-PUFAs and adiposity, but results have been inconsistent. A study in 124 adults living in the UK found that the plasma levels of n-3 PUFA were negatively associated with anthropometric measures of obesity [75], whereas positive associations were reported in a study of 79,839 women living in the US [76]. However, no effect of n-3 LC-PUFA consumption on BMI was found in a 12 year follow-up US cohort of 43,671 men [77]. In a randomised controlled trial (RCT) of 27 women, the intake of a 3 g/d of fish oil (1.8 g n−3 PUFAs) for 2 months was associated with adiposity reduction [78]. Similar findings were reported in an RCT of 324 men and women from Iceland, Spain and Ireland, in which the intake of either lean fish (3 × 150 g portions of cod/week) or fatty fish (3 × 150 g portions of salmon/week), or fish oil (docosahexaenoic acid/eicosapentaenoic acid capsules) for 8 weeks were associated with weight loss in men [79]. However, a 6 week RCT in 195 UK adults found no differences in the anthropometric measures between three intervention diets of specific fatty acid compositions of total energy intake (TEI) (%TEI SFA:%TEI MUFA:%TEI omega-6 PUFA): SFA-rich diet (17:11:4), MUFA-rich diet (9:19:4) or omega-6 PUFA-rich diet (9:13:10) [80]. A meta-analysis of 534,906 European individuals revealed that the higher adherence to the Mediterranean diet, which is rich in MUFA, was associated with a beneficial effect on WC [81]. However, a recent 4 week intervention found no significant effect of the intake of 50 g/day of olive oil, which is rich in MUFA, on BMI or central obesity in 91 UK adults [82]. Conflicting evidence exists regarding the effects of dietary fat on obesity-related traits; this could be because of the genetic heterogeneity and the gene–diet interactions that vary across multiple ethnic groups [83]; hence, the influence of both genetic and lifestyle factors should be considered in understanding the pathophysiology of obesity.

In 2018, the WHO recommended that the intake of total fat and SFA should not exceed 30% and 10% of TEI, respectively, to avoid weight gain [84]. According to the WHO, the recommended range for PUFA for the general population is 6–11% of TEI [85]. It has been identified that the average consumption of SFA in Africa is between 8.9% and 12.5% TEI (North: 9.1%, Central: 12.2, Eastern: 10.7%, Southern: 8.9% and Western Africa: 12.5%; which is slightly higher than the ≤10% TEI recommended by the WHO). The intake of PUFA is low in many sub-Saharan African countries, suggesting the infrequent use of vegetable oils for cooking or preparing foods [86]. The extremely low intake of n-3 long chain PUFA was also identified in Africa, which is explained by the low availability of fish in sub-Saharan Africa countries [86]. In the present study, the average consumption of total fat intake was 23.04 ± 9.13% of TEI and the average consumption of SFA, MUFA and PUFA were 8.95 ± 4.10, 9.86 ± 3.65 and 4.99 ± 1.61% of TEI, respectively, which are in accordance with general dietary recommendations. However, nearly one third of the study population had a high consumption of total fat (mean intake: 34.99 ± 5.54 g/day), the group in which the GRS showed a significant association with a higher WC. Hence, our study suggests that following the general dietary recommendations might be an effective way to overcome the genetic susceptibility to central obesity.

The strengths of our study include the analysis of gene–lifestyle interactions in a well characterized Ghanaian population and the use of different metabolic GRSs to maximise statistical power and to reduce multiple testing [25,27]. Nevertheless, some limitations need to be acknowledged. First, our analysis included an only Ghanaian population, which limits the generalisability of our results to other population groups within Africa. Second, our metabolic GRSs were constructed based on BMI-associated loci predominantly identified in Europeans, which might not truly reflect the genetic associations with BMI among Africans. Third, the food intakes were assessed using repeated 24 h dietary recall method, which is prone to reporting bias and this might have contributed to the discrepancy in the caloric consumption between overweight/obese and non-obese groups [87]. Fourth, as with any cross-sectional study design, residual confounding might exist, despite adjustments for several confounding factors. Fifth, our sample size was small; however, our study had sufficient statistical power to detect significant gene–diet interactions.

## 5. Conclusions

In conclusion, our study has shown that higher intakes of total fat, SFA, MUFA and PUFA can increase the genetic risk on WC in Ghanaian adults. We found that the effect of metabolic risk alleles on WC is stronger among individuals with higher intakes of total fat, SFA, MUFA, PUFA. These results give important insights into the complex interactions between dietary intake and the genetic predisposition to central obesity and highlight the importance of personalising dietary advice according to each ethnic group. Our GRS approach provides insights into cumulative genetic susceptibility; however, studies with a large sample size will be needed to confirm the findings before public health recommendations and personalized nutrition advice can be developed for the Ghanaian population.

## Figures and Tables

**Figure 1 nutrients-12-01906-f001:**
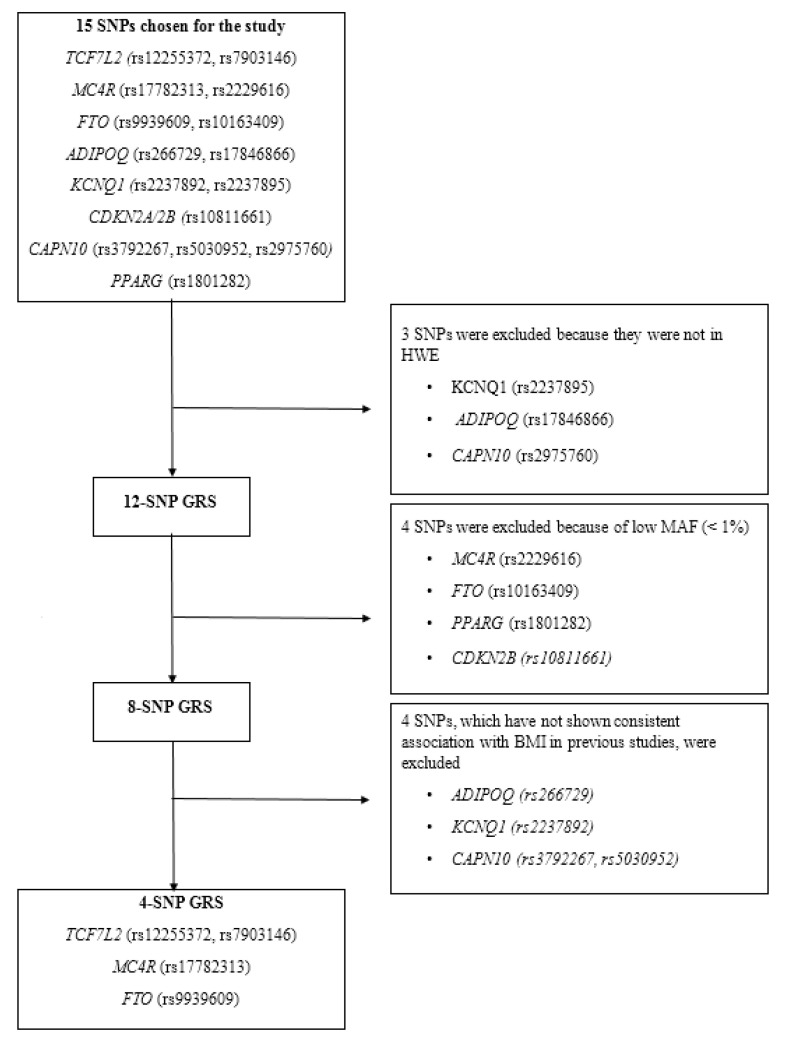
Steps involved in the construction of the metabolic GRS. Fifteen SNPs were genotyped in our study; however, the GRS analysis was based only on 12 SNPs as 3 SNPs were not in the HWE. Three different GRSs, including the 12-SNP GRS, 8-SNP GRS and the 4-SNP GRS were constructed. In the 12-SNP GRS analysis, no significant results were identified, which could be because 4 of the SNPs had MAF of less than 5%. Therefore, the 4 SNPs were excluded, and an 8-SNP GRS was created. No significant findings were observed using the 8-SNP GRS; this could be because four of the eight SNPs have not shown a consistent association with obesity-related traits in other populations. Hence, these four SNPs were removed and a 4-SNP GRS including those SNPs that have shown consistent associations with obesity across several populations was constructed. Abbreviations: SNP: single nucleotide polymorphisms; GRS: genetic risk score; HWE: Hardy–Weinberg equilibrium; MAF: minor allele frequency.

**Figure 2 nutrients-12-01906-f002:**
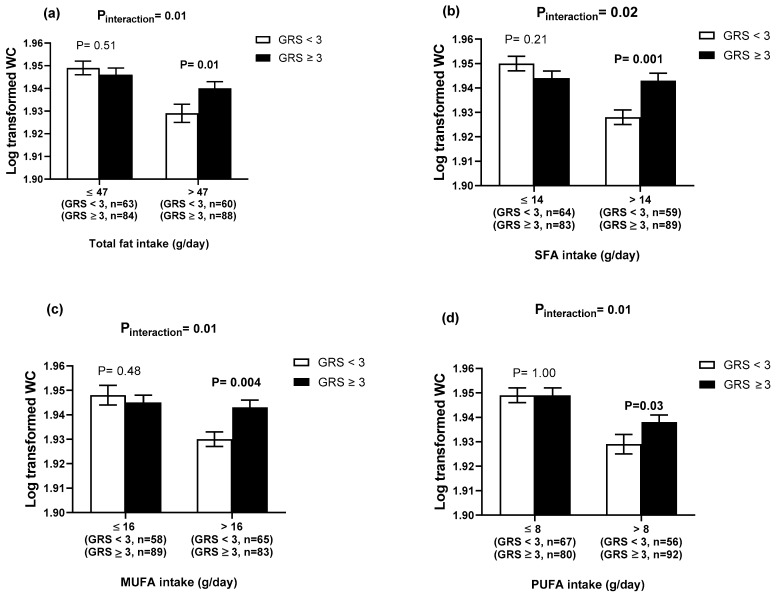
Interaction between the 4-SNP GRS and fat intake (g/day) on the log transformed WC. (**a**) Interaction between the 4-SNP GRS and the log transformed total fat intake (g/day) on WC. White bars indicate individuals with a GRS < 3 risk alleles; the black bars indicate individuals with GRS ≥ 3 risk alleles. Individuals with ≥3 risk alleles had a significantly higher WC compared to those with <3 risk alleles, among individuals with a higher total fat intake (above median group > 47 g/day): 71.28 ± 23.68 g/day (34.99 ± 5.54 % TEI); (**b**) the interaction between the 4-SNP GRS and the log transformed SFA intake (g/day) on the log transformed WC. White bars indicate individuals with a GRS < 3 risk alleles; the black bars indicate individuals with GRS ≥ 3 risk alleles. Individuals with ≥3 risk alleles had a significantly higher WC compared to those with <3 risk alleles, among individuals with a higher SFA intake: 23.50 ± 10.08 g/day (12.19 ± 3.21% TEI); (**c**) the interaction between the 4-SNP GRS and the log transformed MUFA intake (g/day) on the log transformed WC. White bars indicate individuals with a GRS < 3 risk alleles; the black bars indicate individuals with GRS ≥ 3 risk alleles. Individuals with ≥3 risk alleles had a significantly higher WC compared to those with <3 risk alleles, among individuals with a higher MUFA intake: 25.72 ± 9.58 g/day (12.79 ± 2.53% TEI); (**d**) the interaction between the 4-SNP GRS and the log transformed PUFA intake (g/day) on the log transformed WC. White bars indicate individuals with a GRS < 3 risk alleles; the black bars indicate individuals with GRS ≥ 3 risk alleles. Individuals with ≥3 risk alleles had a significantly higher WC compared to those with <3 risk alleles, among individuals with a higher PUFA intake: 12.74 ± 4.7 g/day (6.28 ± 1.08% TEI). Abbreviations: SNP: single nucleotide polymorphisms; GRS: genetic risk score; WC: waist circumference; SFA: saturated fatty acids; MUFA: monounsaturated fatty acids; PUFA: polyunsaturated fatty acids; TEI: total energy intake. Error bars indicate the standard error of the mean.

**Table 1 nutrients-12-01906-t001:** Characteristics of the study participants.

	Total (N = 302)	Non-Obese * (N = 126)	Overweight/Obese ** (N = 176)	*p* Value ***
Age (years)	38.17 ± 9.64	35.96 ± 9.55	39.75 ± 9.42	0.001
BMI (kg/m^2^)	26.63 ± 4.99	22.01 ± 1.79	29.95 ± 3.75	<0.001
WC (cm)	88.48 ± 12.41	77.99 ± 7.13	96.00 ± 9.61	<0.001
WHR	1.45 ± 6.96	1.55 ± 7.76	1.38 ± 6.34	0.84
Visceral fat (%)	8.02 ± 7.39	6.49 ± 10.97	9.12 ± 2.26	0.01
Body fat (%)	33.12 ± 13.90	22.05 ± 12.47	41.05 ± 8.36	<0.001
Total energy intake (%)	1647.93 ± 685.83	1772.17 ± 723.85	1558.99 ± 644.75	0.008
Protein intake (g/day)	53.24 ± 23.73	57.38 ± 24.52	50.28 ± 22.76	0.01
Total fat intake (g/day)	51.17 ± 26.94	55.00 ± 29.29	48.42 ± 24.85	0.04
Carbohydrates intake (g/day)	239.03 ± 95.84	259.44 ± 104.01	224.42 ± 86.94	0.002
Fibre intake (g/day)	21.31 ± 10.84	23.19 ± 11.44	19.96 ± 10.21	0.01
Total SFA intake (g/day)	16.23 ± 10.36	17.41 ± 11.29	15.39 ± 9.58	0.10
Total MUFA intake (g/day)	18.08 ± 10.49	19.63 ± 11.30	16.96 ± 9.74	0.03
Total PUFA intake (g/day)	9.12 ± 5.03	10.20 ± 5.56	8.35 ± 4.47	0.002

Data presented as the means ± standard deviations. * Non-obese individuals refer to the individuals with a BMI < 25 Kg/m^2^, according to the WHO classification of BMI. ** Overweight/obese cases refer to individuals with BMI ≥ 25 Kg/m^2^, according to the WHO classification of BMI. *** *p* values for the differences in the means between the two groups were calculated using the independent samples t-test. Abbreviations: BMI: body mass index; WC: waist circumference; WHR: waist–hip ratio; SFA: saturated fatty acids; MUFA: monounsaturated fatty acids; PUFA: polyunsaturated fatty acids; WHO: World Health Organisation.

**Table 2 nutrients-12-01906-t002:** Associations of the 4-SNP GRS on obesity-related traits.

	GRS < 3 Risk Alleles (N = 123)	GRS ≥ 3 Risk Allele (N = 172)	* *p* Value
BMI (kg/m^2^)	26.13 ± 0.45	26.85 ± 0.37	0.24
WC (cm)	87.13 ± 1.15	89.14 ± 0.92	0.19
WHR	2.27 ± 0.98	0.88 ± 0.01	0.18
Visceral fat (%)	7.89 ± 0.71	8.08 ± 0.55	0.43
Body fat (%)	31.75 ± 1.32	33.87 ± 1.02	0.15

* *p* Values obtained from the linear regression analysis adjusted for age, sex and additionally for BMI when BMI is not an outcome. The analysis was performed on log-transformed variables. Abbreviations: SNP: single nucleotide polymorphism; GRS: genetic risk score; BMI: body mass index; WC: waist circumference; WHR: waist–hip ratio.

**Table 3 nutrients-12-01906-t003:** Interactions between the 4-SNP GRS and the lifestyle factors on obesity-related traits.

	Protein (g/day)	Carbohydrate (g/day)	Fibre (g/day)	Fat (g/day)	SFA (g/day)	MUFA (g/day)	PUFA (g/day)	Physical Activity
**BMI (kg/m^2^)**	0.45	0.22	0.12	0.15	-	-	-	0.76
**WC (cm)**	0.08	0.21	0.41	0.01	0.02	0.01	0.01	0.24
**WHR**	0.82	0.88	0.49	0.80	-	-	-	0.002
**Visceral fat (%)**	0.50	0.35	0.32	0.38	-	-	-	0.93
**Body fat (%)**	0.46	0.11	0.04	0.75	-	-	-	0.60

Data are *p* values obtained from the linear regression analysis adjusted for age, sex, total energy intake and additionally for BMI when BMI is not an outcome. The analysis was performed on log-transformed variables. Abbreviations: SNP: single nucleotide polymorphism; GRS: genetic risk score; BMI: body mass index; WC: waist circumference; WHR: waist–hip ratio; SFA: saturated fatty acids; MUFA: monounsaturated fatty acids; PUFA: polyunsaturated fatty acids.

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
