# Peer review of "Interaction between Metabolic Genetic Risk Score and Dietary Fatty Acid Intake on Central Obesity in a Ghanaian Population"

_nutrients, 2020, doi:10.3390/nu12071906_

Round 1

Reviewer 1 Report

Comments to the Author

In this manuscript, the authors have analyzed the interaction between genetic SNPs and lifestyle factors on obesity-related traits in 302 healthy Ghanaian adults. Further, they suggested that higher consumption of dietary fatty acids may have the potential to increase the genetic susceptibility of becoming centrally obese. Further, they present a general nutritional recommendation to decrease the intake of total fat and saturated fatty acids, to reduce the risk of obesity with a genetic predisposition to central obesity. The results are of interest, and the manuscript is overall well written. However, there are some concerns which should be addressed

  1. Introduction: It is good to give importance to other factors environmental, nutritional, epigenetic interactions of obesity etiology beyond SNPs from preclinical animal studies or human studies. It is suggestive to include few recent studies (PMID: 31616013, PMID:31610854, PMID:25232145, PMID: 32067051, PMID: 21466928) on those aspects (though few given in discussion).
  2. Figure 1. A total of 15 SNPs chosen for the study; finally, 4 SNPs, which have not shown a consistent association with BMI in previous studies, were excluded. “Then why authors chose those; did authors found any association with metabolic conditions in this particular population.”
  3. Figure 2: Is that Error bar, SEM, or SD; it must be given in the legend.
  4. Even though gender (M: F) classifications given in supplementary data, it is good to mention subject characterization in the main manuscript table also. Did the authors found any gender bias between metabolic-GRS? And it must be discussed.
  5. Limitations/caveat of the study must include before the conclusion paragraph.
  6. What was the unique feature seen in the Ghanaian population; for a comparison with other studies of the replica.
  7. There are many other minor errors of syntax and grammar throughout the text, which need to be fixed. Check abbreviations in the first instances throughout the manuscript.

Author Response

Responses to the reviewer’s comments

We thank the reviewers for their comments for the manuscript entitled ‘Interaction between metabolic-genetic risk score and dietary fatty acid intake on central obesity in a Ghanaian population’ (Manuscript ID: nutrients-779258). We have responded to each of their comments below and included the track changes (highlighted in yellow) in the marked manuscript.

Responses to the comments from Reviewer 1:

In this manuscript, the authors have analyzed the interaction between genetic SNPs and lifestyle factors on obesity-related traits in 302 healthy Ghanaian adults. Further, they suggested that higher consumption of dietary fatty acids may have the potential to increase the genetic susceptibility of becoming centrally obese. Further, they present a general nutritional recommendation to decrease the intake of total fat and saturated fatty acids, to reduce the risk of obesity with a genetic predisposition to central obesity. The results are of interest, and the manuscript is overall well written. However, there are some concerns which should be addressed

We thank the reviewer for the feedback.

  1. Introduction: It is good to give importance to other factors environmental, nutritional, epigenetic interactions of obesity etiology beyond SNPs from preclinical animal studies or human studies. It is suggestive to include few recent studies (PMID: 31616013, PMID:31610854, PMID:25232145, PMID: 32067051, PMID: 21466928) on those aspects (though few given in discussion).

As suggested by the reviewer, we have now modified the introduction and included the suggested references on Pages 1-2, lines 44-45.    

  1. Figure 1. A total of 15 SNPs chosen for the study; finally, 4 SNPs, which have not shown a consistent association with BMI in previous studies, were excluded. “Then why authors chose those; did authors found any association with metabolic conditions in this particular population.”

We had selected 15 SNPs based on the previous genome-wide association studies for metabolic traits [1-7]. Of these 15, three were not in Hardy-Weinberg equilibrium and the remaining 12 SNPs as a genetic risk score (GRS) did not show any significant interaction with lifestyle factors (Supplementary Table 3). We further excluded four SNPs which had a minor allele frequency <5%. The 8-SNP GRS also did not show any significant interaction (Supplementary Table 4). Finally, after excluding four SNPs, which did not show a consistent association with BMI in other populations, the remaining 4-SNP GRS showed a significant interaction (Table 2). Even though 15 SNPs were chosen in the first step, we had to exclude additional SNPs based on the above-mentioned criteria to create an appropriate GRS for the present nutrigenetics study. The steps involved in the construction of the 4-SNP GRS are shown in Figure 1.

References: 1. Berndt, 2013; 2. Locke et al, 2015; 3.Okada et al, 2012; 4.Speliotes et al, 2010; 5.Wen et al, 2012; 6.Willer et al, 2010; 7.Winkler et al, 2016.

  1. Figure 2: Is that Error bar, SEM, or SD; it must be given in the legend.

The error bars in the Figure 2 are SEMs. We have now included this information in the legend of Figure 2 (Page 10, lines 263-264).

  1. Even though gender (M: F) classifications given in supplementary data, it is good to mention subject characterization in the main manuscript table also. Did the authors found any gender bias between metabolic-GRS? And it must be discussed.

As suggested by the reviewer, we have included the frequency of women and men among obese and non-obese in Table 1. In our study, we did not find any effect of gender on the main and interaction effects. All the statistical analyses were adjusted for gender. In addition, we have also tested for the interaction between 4-SNP GRS and gender on obesity-related traits and no significant interactions with gender were detected (p> 0.05) (Page 7, Line 241-243).

  1. Limitations/caveat of the study must include before the conclusion paragraph.

The strengths and limitations are included before the Conclusion section on pages 12-13, lines 368-379.

  1. What was the unique feature seen in the Ghanaian population; for a comparison with other studies of the replica.

In our Ghanaian population, we found that women had significantly higher levels of BMI, body fat percentage and waist-hip ratio compared to men, despite men consuming significantly higher levels of carbohydrate, protein and fat (P<0.05). Previous reviews have also observed a higher prevalence of metabolic syndrome (MetS) among Ghanaian women than men [1]. Additionally, the prevalence of MetS was almost four times higher in Ghanaian women compared to men and these differences were suggested to be driven by the significant and higher prevalence of overweight and obesity among Ghanaian women (55.0%) compared to men (29.0%) [2]. Hence, the observed finding in our study could be an unique feature of Ghanaian population.

Countries in sub-Saharan Africa are clearly undergoing a nutrition transition [3]. Our study was conducted in Kumasi, which is the capital of Ghana’s Ashanti region and one of the largest and fastest-growing urban areas in the country which makes it quite unique compared to other African countries. An increase in fat consumption has been shown to be highly associated with global dietary changes [3] and, interestingly, our findings were mainly restricted to the interaction with dietary fatty acid intake, which is in line with the findings in European populations [4,5]. 

Our study is the first study of its kind, investigating the effect of different metabolic-GRSs on obesity traits in a Ghanaian population. We found that none of the metabolic-GRSs was significantly associated with obesity-related traits, which contradicts the findings of the previous GRS-related studies in European and African populations [6-14]. Even though this finding is unique to Ghanaian population, previously reported genetic associations of individual SNPs with obesity measures in non-African populations have also shown only limited success among Africans [15-18].

References: 1.Ofori-Asenso et al, 2017; 2.Gyakobo et al, 2012; 3.Steyn et al, 2014; 4.Celis-Morales et al, 2017; 5.Casas-Agustench et al, 2014; 6.Speliotes et al, 2010; 7.Hung et al, 2015; 8.Sandholt et al, 2010; 9.Li et al, 2010; 10.Badsi et al, 2014; 11.Fulford et al, 2015; 12.Munthali et al, 2018; 13.Belsky et al, 2013; 14.Peterson et al, 2011; 15.Adeyemo et al, 2010; 16.Hennig et al, 2009; 17.Logan et al, 2016; 18.Sahibdeen et al, 2018.

  1. There are many other minor errors of syntax and grammar throughout the text, which need to be fixed. Check abbreviations in the first instances throughout the manuscript.

We have corrected the typos and grammatical errors and abbreviated the words in the first instance, wherever appropriate.  

Reviewer 2 Report

This study describes the relationship between metaboli-genetic risk score and dietary FFA on waist circumference in Ghanaian adults.

Major problems:

1.- The size of the population is small for a cross-sectional study.

2.- Looking at table S2, it can be found that men population weren´t obesity, even overweight, however women sample were overweight/obese. Thus, the population of the sample was not homogeneous. This complicates data analysis and conclusions.

3.- In table S2, it is observed that the amount of visceral fat is the same between men and women, however, the amount of total body fat is double in women than in men. This data are difficult to understand.

4.- In a cross-sectional study, it can only be established an association between the intakes of total fat and the SNPs studied, but it can not be concluded that the intake of higher amounts of fat is the consequence of a bigger WC in the population studied. To establish a relationship longitudinal studies are needed.

5.- The major novelty of the study is that it has been performed in an African population. In this regard, as the authors indicate, there exist many studies arriving to the same conclusions in European populations. 

Author Response

Responses to the reviewer’s comments

We thank the reviewers for their comments for the manuscript entitled ‘Interaction between metabolic-genetic risk score and dietary fatty acid intake on central obesity in a Ghanaian population’ (Manuscript ID: nutrients-779258). We have responded to each of their comments below and included the track changes (highlighted in yellow) in the marked manuscript.

Responses to the comments from Reviewer 2:

This study describes the relationship between metabolic-genetic risk score and dietary FFA on waist circumference in Ghanaian adults.

Major problems:

1.- The size of the population is small for a cross-sectional study.

We agree with the reviewer that the sample size is small; however, despite the small sample size, we were still able to identify significant gene-diet interactions, which suggest that our study has sufficient statistical power. We have also mentioned the small sample size as a limitation on Page 13, lines 378-379.

  1. Looking at table S2, it can be found that men population weren´t obesity, even overweight, however women sample were overweight/obese. Thus, the population of the sample was not homogeneous. This complicates data analysis and conclusions.

As mentioned in the response to the 6th comment from reviewer 1, women had significantly higher levels of BMI, body fat percentage and waist-hip ratio compared to men, despite men consuming significantly higher levels of carbohydrate, protein and fat. This was a finding of the present study and not a biased selection of men and women. As listed on Page 2, lines 82-86, under ‘Study Population’, five communities were selected randomly from the list of communities in the Oforikrom Municipal Assembly. In each community, a central point was located (a vehicle station, marketplace or other landmarks) and a fieldworker entered the first house that faced either North, South, East or West of that central point and randomly recruited one respondent from each household. Furthermore, all our analyses were adjusted for gender and GRS-gender interactions were not significant as well. Hence, our study population was randomly chosen and hence our findings are unlikely to be affected by the differences between men and women.

3.- In table S2, it is observed that the amount of visceral fat is the same between men and women, however, the amount of total body fat is double in women than in men. This data are difficult to understand.

As mentioned in the response to the 6th comment from reviewer 1, there are a few unique features of Ghanaian population that were identified in the present study and one such finding is the difference in obesity traits between men and women. But these findings have already been observed in previous studies. Previous reviews [1] have shown a higher prevalence of metabolic syndrome among Ghanaian women than men. Additionally, the prevalence of metabolic syndrome was almost four times higher in Ghanaian women compared to men; these differences were suggested to be driven by the higher prevalence of overweight and obesity among women (55.0%) compared to men (29.0%) [2]. These findings make the population unique and interesting compared to other African counterparts.

References: 1. Ofori-Asenso et al, 2017; 2. Gyakobo et al, 2012

4.- In a cross-sectional study, it can only be established an association between the intakes of total fat and the SNPs studied, but it can not be concluded that the intake of higher amounts of fat is the consequence of a bigger WC in the population studied. To establish a relationship longitudinal studies are needed.

In the present study, we have not shown any causal effect and nowhere in the text have we used the word ‘causal’. As mentioned on Page 13, lines 381-383, under ‘Conclusion’, our study has shown that higher intakes of total fat, SFA, MUFA and PUFA can increase the genetic risk on WC in Ghanaian adults. We found that the effect of metabolic risk alleles on WC is stronger among individuals with higher intakes of total fat, SFA, MUFA, PUFA. This clearly indicates that the effects are not causal.

  1. The major novelty of the study is that it has been performed in an African population. In this regard, as the authors indicate, there exist many studies arriving to the same conclusions in European populations. Interaction with fat what is required to be done in this population

Although genetic research in Africans is increasing in numbers [1], only a few studies have examined the association of GRS with obesity in Africa and no studies have tested interactions between GRS and lifestyle factors on obesity [2-4], which highlights the need of further research in African populations. GRS-lifestyle interaction studies have largely been conducted in populations of European ancestry [5,6]. In agreement with our study, data from UK Biobank [7] and two studies from the US [8] have reported significant interactions between GRS and dietary intakes of total fat and SFA on WC; the GRS was associated with higher WC among individuals with high intakes of total fat and SFA. However, interactions on BMI were not identified in the present study, which contradicts the previously reported findings [7,8]. Hence, larger studies are required to replicate our GRS-fat intake interactions on WC in the Ghanaian population.

References: 1.Rotimi et al, 2017; 2.Badsi et al, 2014; 3.Fulford et al, 2015; 4.Munthali et al, 2018; 5.Kilpelainen et al, 2011; 6.Qi et al, 2012; 7.Celis-Morales et al, 2017; 8.Casas-Agustench et al, 2014.